# Comparison of Malondialdehyde, Acetylcholinesterase, and Apoptosis-Related Markers in the Cortex and Hippocampus of Cognitively Dysfunctional Mice Induced by Scopolamine

**DOI:** 10.3390/biomedicines12112475

**Published:** 2024-10-28

**Authors:** Hee-Jung Park, Myeong-Hyun Nam, Ji-Hoon Park, Ji-Min Lee, Hye-Sun Hong, Tae-Woo Kim, In-Ho Lee, Chang-Ho Shin, Soo-Hong Lee, Young-Kwon Seo

**Affiliations:** 1Department of Biomedical Engineering, Dongguk University, Goyang-si 10326, Republic of Korea; 2Department of AI Convergence Biomedical Engineering, Dongguk University, Goyang-si 10326, Republic of Korea; 3AriBio Co., Ltd., Seongnam-si 13535, Republic of Korea

**Keywords:** cognitive dysfunction, scopolamine, cortex, hippocampus

## Abstract

**Objectives:** Until now, many researchers have conducted evaluations on hippocampi for analyses of cognitive dysfunction models using scopolamine. However, depending on the purposes of these analyses, there are differences in the experimental results for the hippocampi and cortexes. Therefore, this study intends to compare various analyses of cognitive dysfunction after scopolamine administration with each other in hippocampi and cortexes. **Methods:** Scopolamine was administered at three dosages in mice: 0.5, 1, and 3 mg/kg. And this study evaluates the differences in cognitive function and the expression of malondialdehyde (MDA), acetylcholinesterase (AChE), and brain-derived neurotrophic factor (BDNF) in mice’s hippocampi and cortexes based on scopolamine dosages. **Results:** The Morris water maze test was conducted between 1 and 3 h after scopolamine injection to assess its duration. A significant decrease in behavioral ability was evaluated at 1 h, and we observed a similar recovery to the normal group at 3 h. And the Morris water maze escape latency showed differences depending on scopolamine concentration. While the escape waiting time in the control group and scop 0.5 administration group remained similar to that seen before administration, the administration of scop 1 and 3 increased it. In the experimental group administered scop 1 and 3, cerebral MDA levels in the cerebral cortex significantly increased. In the hippocampus, the MDA level in the scopolamine-administered groups slightly increased compared to the cortex. A Western blotting assay shows that Bax and Bcl-xl showed a tendency to increase or decrease depending on the concentration, but BDNF increased in scop 0.5, and scop 1 and 3 did not show a significant decrease compared to the control at the cerebral cortex. In the hippocampus, BDNF showed a concentration-dependent decrease in expression. **Conclusions:** This study’s findings indicate that chemical analyses for MDA and AChE can be performed in the cerebral cortex, while the hippocampus is better suited for protein analysis of apoptosis and BDNF.

## 1. Introduction

Neurodegenerative diseases are chronic conditions that impair or destroy the nervous system, especially the brain, over time. Most of these conditions are closely related to age [1,2].

Alzheimer’s disease (AD) is a major neurodegenerative disorder that presents with progressive memory loss and cognitive decline, finally leading to respiratory distress and death [3,4]. The pathological characteristic of AD is amyloid-beta (Aβ) plaque formation [5,6]. Accumulation of Aβ plaques causes the degeneration and death of cholinergic neurons, resulting in synapse loss, which causes cognitive dysfunction and, ultimately, death [3,4,7,8,9]. Therefore, acetylcholine plays an integral role in the treatment and alleviation of AD [10,11,12,13,14]. Computer simulations have explored many new drug candidates over the past few years, but the clinical development stage has confirmed their ineffectiveness [15,16]. Selecting an animal model suitable for the new drug development process is an important step in the process [17]. Since AD involves several interconnected molecular pathways, appropriate animal models must be created to understand the underlying mechanisms and develop new drugs [6,18,19]. Cognitive decline, behavioral imbalance, deterioration of neuronal function, neuronal loss, and oxidative stress are common in AD caused by aging or drugs [20,21,22]. Also, many studies have shown that BDNF plays a crucial role in neurodegenerative disorders [23,24,25,26,27].

Scopolamine is a competitive antagonist of muscarinic acetylcholine receptors that blocks cholinergic neurotransmission and causes memory impairment in rodents [28,29,30]. Recent studies have shown that scopolamine causes oxidative stress, leading to increases in reactive oxygen species and memory impairment [31,32]. Additionally, scopolamine can induce apoptosis in hippocampus neuronal cells [33,34]. The cholinergic hypothesis may be applicable, with scopolamine injections causing cognitive and memory deficits like those observed in AD.

Additionally, scopolamine has been known to precisely replicate a condition that, while not as severe as classic dementias, manifests as acute Transient Global Amnesia (TGA). This is a benign and transient form of anterograde amnesia that typically resolves within 24 h, and is induced by the muscarinic deficit influencing central noradrenergic tones [35].

Through the development of transgenic mouse models, AD models such as the 5X FAD model and the APP/PS1 model have been developed, but their prices are high. In particular, there is a problem that some models show normal levels in exercise performance evaluations, although transgenicity is confirmed. Therefore, it can be a financial burden in experiments evaluating many animals.

When creating a specific disease model, the clinical phenotype characteristics must be considered. In related studies that created a cognitive dysfunction model using scopolamine, various doses have been administered [36,37,38,39]. These different doses of scopolamine show differences in model formation, which can lead to different research results. Related researchers compared single and repeated administrations of scopolamine at doses of 0.5, 1, and 3 mg/kg, and compared the cognitive decline resulting from this through a memory retrieval test. In addition, hippocampus tissue was analyzed for AChE and MDA. As a result, it was reported that AChE and MDA increased with single administration, and the effect increased further with repeated administration [40].

Analyses of cognitive dysfunction require mRNA and protein analyses for various markers in addition to chemical analyses. Many researchers have conducted evaluations targeting the hippocampus for analysis. However, there has been no study that performed a cross-comparison evaluation of the hippocampus and cortex for these various analyses. Therefore, this study aims to perform a cross-comparison of various analyses of cognitive dysfunction after scopolamine administration in hippocampi and cortexes.

We aimed to conduct research on cognitive dysfunction based on changes in exploratory behavior (i.e., novel object recognition test), learning and memory behavior (i.e., Morris water maze), and cholinergic system in a new environment according to the doses of scopolamine (0.5, 1, and 3 mg/kg) and analysis biomarkers related to cognitive function, such as apoptotic markers, MDA levels, AChE activity, and BDNF expression in the hippocampus and cortex.

## 2. Materials and Methods

### 2.1. Animals

Male ICR mice (weighing 20–25 g, aged 4 weeks) were purchased from Orient Bio with approval from the Animal Research Ethics Committee. The experimental protocol was approved by the Institutional Animal Care and Use Committee of Dongguk University (IACUC-2022-016-2). They housed 4–5 mice per cage at room temperature (22 °C ± 2 °C) with a 12 h shifting light–dark cycle, and food and water were provided freely. Drug treatment started 7 days after the animals arrived.

### 2.2. Drugs and Experimental Protocol

#### 2.2.1. Evaluation of Scopolamine Duration

Scopolamine hydrobromide was purchased from Sigma Aldrich (S0929). Before the experiments, sterilized 0.9% saline solution was used to dilute scopolamine. Diluted scopolamine was administered intraperitoneally.

To evaluate the persistence of scopolamine, the Morris water maze test was conducted between 1 h and 3 h after the administration of scopolamine.

Briefly, the Morris water maze test was conducted in a water-filled circular open pool, the temperature of which was sustained at 22 °C ± 1 °C. The surface of the water was partitioned into four quadrants (Q1, Q2, Q3, and Q4) by two primary vertical axes. The test was carried out on the final day following saline or SCO treatment, with training conducted over three consecutive days prior to the test (saline and SCO were administered continuously during the training and testing periods). The mice swam for 120 s to find the hidden platform. If the platform was not found within 120 s, the escape was assisted, and the escape delay was recorded as 120 s. At the end of each trial, the mouse stayed on the platform for 30 s. The ANY-maze video tracking system (Version 6.3, Stoelting Co., Wood Dale, IL, USA) was used to record escape latency (the time taken to find platform).

#### 2.2.2. Design of Scopolamine Dose Experiment

Scopolamine was set at three doses: 0.5, 1, and 3 mg/kg. The four groups were the following: (1) saline (control), (2) 0.5 mg/kg scopolamine (scop 0.5), (3) 1 mg/kg scopolamine (scop 1), and (4) 3 mg/kg scopolamine (scop 3). The mice conducted the behavior tests 1 h after intraperitoneal scopolamine injection. All mice performed the Morris water maze test and the novel object recognition test. One day after the end of the preceding behavioral experiments, the mice were administered scopolamine before being sacrificed for biochemical analyses, Western blot, and immunohistochemistry (IHC).

### 2.3. Behavior Test

#### 2.3.1. Morris Water Maze

After grouping according to the dose of scopolamine administered to the mice, a Morris water maze test was conducted.

The Morris water maze test was conducted in a water-filled circular open pool, the temperature of which was sustained at 22 °C ± 1 °C. The surface of the water was partitioned into four quadrants (Q1, Q2, Q3, and Q4) by two primary vertical axes. Visual signs were attached to the walls of each quadrant. The ANY-maze video tracking system was used to record swimming paths, and the data were analyzed. Each day, animals started in a random quadrant and underwent the experiment. The platform in the center of the Q1 platform was submerged 1 cm below the water’s surface. During the acquisition phase for 4 days, the mice swam for 120 s to find the hidden platform. If the platform was not found within 120 s, the escape was assisted. At the end of each trial, the mouse stayed on the platform for 30 s. A memory test was performed 24 h after the last training session. The platform was then eliminated from the pool and the mice were tested for 120 s. Among other parameters, the time spent on the platform quadrant (Q1) and the platform were measured.

#### 2.3.2. Novel Object Recognition Test

The experimental setup involved an open field made of polyvinyl plastic, measuring 25 cm on each side (25 cm × 25 cm × 25 cm). During habituation training, the animals were placed in the apparatus for 10 min each day without any objects present. During the test phase, mice were introduced to the experimental apparatus containing two identical objects and were allowed to explore for 10 min. Four hours later, one of the original objects was replaced with a novel one, and the mice were reintroduced to the apparatus. Again, they were given 10 min to explore. The objects selected for the experiment were of similar height. The time spent by the mice exploring each object was recorded, noting the duration spent with the familiar object (N1) and the novel object (N2).

### 2.4. Lipid Peroxidation

Lipid peroxidation within tissues can be used to assess oxidative stress. Malondialdehyde (MDA), a major marker of lipid peroxidation, was measured in the homogenized tissue samples using a colorimetric lipid peroxidation assay kit, following the provided protocol. MDA in the sample reacts with thiobarbituric acid (TBA) to form MDA-TBA adducts, which were detected colorimetrically at 540 nm.

### 2.5. AChE Activity

Acetylcholinesterase (AChE) is responsible for catalyzing the hydrolysis of acetylcholine, a neurotransmitter involved in neurosignaling. AChE activity in hippocampal and cerebral cortex homogenates was measured using an AChE activity assay kit, following the manufacturer’s instructions. In short, acetylcholine is broken down into choline by AChE, and the choline is then oxidized by choline oxidase, producing H_2_O_2_**.** This reaction leads to color development in the probe, which was measured at 570 nm.

### 2.6. Western Blotting

Protein levels were analyzed using Western blotting. Proteins were extracted using previously established methods, and their concentrations were measured using the bicinchoninic acid assay. In total, 50 μg of protein from each sample was loaded onto a 10% ShDS-PAGE gel, run at 90 V for 120 min, and transferred onto nitrocellulose membranes. The membranes were blocked with 5% skimmed milk for 1 h at room temperature. After three washes with TBST, they were incubated with primary antibodies targeting Bax, Bcl-XL, and BDNF for 1 h. Following additional washing, the membranes were exposed to an anti-rabbit secondary antibody (2 μg) in 5% skimmed milk/TBST for 2 h, followed by further washes. The protein bands were visualized using enhanced chemiluminescence (ECL) and captured on the ChemiDoc XRS+ imaging system (Bio-Rad, Hercules, CA, USA).

### 2.7. Immunohistochemical Analyses of BDNF, Nerve Growth Factor (NGF), and MAP2 Marker Expression

BDNF, NGF, and MAP2 expression levels were evaluated using IHC. The brain tissues were fixed in paraformaldehyde for 24 h and then were dehydrated using a series of graded alcohol solutions and embedded in paraffin. The brain tissue was cut to 4 μm in thickness, and the sections on slides were dried. The slides were baked in a 65 °C oven for 2 h. Anti-BDNF (1:500), anti-NGF (1:100) and anti-MAP2 antibodies (1:200) were incubated at 4 °C overnight. Additionally, goat anti-rabbit HRP-conjugated secondary antibodies (1:500) were applied and incubated at room temperature for 15 min. The DAB chromogenic agent developed color, was rinsed thoroughly with tap water, and was counterstained with hematoxylin for 8 min.

### 2.8. Statistical Analyses

All experiments were performed in triplicate, and the data are presented as mean ± standard error. Differences between groups were analyzed using a one-way analysis of variance (ANOVA) followed by the Dunnett post hoc test. For datasets that did not follow a normal distribution, the non-parametric Kruskal–Wallis test was used. Statistical significance was defined as *p* < 0.05. In the figures, the symbols *, **, and *** indicate *p*-values of less than 0.05, 0.01, and 0.005, respectively.

## 3. Results

### 3.1. Cognitive Impairment Caused by Scopolamine Does Not Persist Long-Term

When compared between 1 h and 3 h after scopolamine administration, Morris water maze escape latency decreased 3 h after administration. In addition, the escape latency was similar to the control group 3 h after scopolamine administration (Figure 1A).

Compared to the escape latency measurement results on the first day, each experimental group decreased by 63.7% (control), 30.6% (1 h after), and 54.6% (3 h after) (Figure 1B). Throughout the period, a decrease in escape latency similar to the control was observed in the curve 3 h after scopolamine administration, whereas the curve 1 h after scopolamine administration showed a smaller reduction in escape latency. It was confirmed that scopolamine administration did not induce cognitive dysfunction 3 h after administration.

### 3.2. Spatial Learning and Memory Impairment Due to Scopolamine Depends on the Dose

We observed the movement paths in the Morris water maze and confirmed that scopolamine administration increased movement to other areas outside the platform. In addition, we confirmed that as the dose of scopolamine increased, movement to other areas increased (Figure 2A).

Morris water maze escape latency showed differences depending on the dose of scopolamine administered. The escape latency times in the control group and scop 0.5 group were similar to those before administration, but the escape latency times of the scop 1 and scop 3 groups increased compared to before administration (Figure 2B).

The exploration time on the quadrant around the platform was measured during an exploration test session. An administered dose of scopolamine caused the time to decrease. The exploration time took upon the quadrant around the platform was similar to before administration in the control group and scop 0.5 group. However, the exploration time took upon the quadrant around platform was found to decrease in the scop 1 and scop 3 groups (Figure 2C).

The virtual platform maintenance time increased in the scop 0.5 group compared to before administration. Additionally, it was confirmed that the virtual platform maintenance time decreased in the scop 1 or scop 3 groups. In particular, a significant decrease was confirmed in the scop 3 group. Therefore, it was confirmed that scopolamine (3 mg/kg) administration can immediately worsen spatial learning and memory (Figure 2D).

### 3.3. Scopolamine-Induced Recognition Memory Impairment Is Reduced at Certain Doses

To determine the effects of scopolamine on learning and cognitive impairment, a novel object recognition test was progressed. In the control group, new object search time significantly increased compared to existing object search time. Scopolamine did not change the difference in exploration time in the scop 0.5 and scop 1 groups. Additionally, the new object exploration time decreased in the scop 3 group. Compared with the control group, the time taken to explore novel objects was significantly reduced in the scop 3 group (Figure 3). Therefore, it was confirmed that the administration of scopolamine (3 mg/kg) could immediately worsen recognition memory because it interfered with the exploration of new objects.

### 3.4. Immunohistochemical Analysis of BDNF, NGF, and MAP2 Marker Expression

To investigate the changes in neurotrophic proteins in the brain following scopolamine administration, the expression levels of BDNF, NGF, and MAP2 were assessed using IHC (Figure 4A–L). In the scop 3 group, BDNF showed limited distribution compared to the control group and was confirmed to be reduced in the hippocampus. Similarly, NGF expression in the hippocampus was weak across all scopolamine-administered groups, with minimal expression observed in the scop 3 group. MAP2 expression was evaluated in the cortex, where the control group exhibited strong microtubule assembly. However, compared to the control group, microtubule assembly was not clearly observed in the scop 0.5 group, and in the scop 1 and 3 groups, the microtubule assembly structure was barely detectable.

The changes in neurotrophic factors according to the dose of scopolamine in cerebral cortex and hippocampal protein were analyzed by Western blotting. In the cerebral cortex, BDNF levels were elevated in the scop 0.5 group, while no significant decrease was observed in the scop 1 and 3 groups compared to the control. However, BDNF expression was found to decrease in a scopolamine-dose-dependent manner in the hippocampus, and BDNF was most significantly reduced in the scop 3 group. (Table 1)

### 3.5. Changes in Malondialdehyde Level in the Brain Following Scopolamine Administration

The group administered scopolamine showed higher MDA levels in the hippocampus and cerebral cortex compared to the control group. The MDA level in the cerebral cortex was different for each experimental group (control: 0.299 μmol/mg; scop 0.5: 0.732 μmol/mg; scop 1: 0.947 μmol/mg; scop 3: 0.983 μmol/mg) (Figure 5A). In the scop 1 and scop 3 groups, the MDA levels in the cerebral cortex increased notably. Additionally, the hippocampus MDA level was 0.157 μmol/mg for the control group, 0.327 μmol/mg for scop 0.5, 0.241 μmol/mg for scop 1, and 0.485 μmol/mg for scop 3 (Figure 5B). Compared with the control group, the MDA levels of both experimental groups increased slightly.

### 3.6. Changes in Acetylcholinesterase (AChE) Activity in the Brain Following Scopolamine Administration

The group administered scopolamine showed increased AChE activity in the hippocampus and cerebral cortex compared to the control group. For each experimental group, AChE activity was assessed in the cerebral cortex, as follows: control group (0.635 nmol/mg), scop 0.5 experimental group (0.805 nmol/mg), scop 1 experimental group (1.160 nmol/mg), and scop 3 experimental group (1.480 nmol/mg). In the experimental group administered scopolamine, AChE activity increased dose-dependently in the cerebral cortex (Figure 6A). In the hippocampus, AChE activity in the control group was 0.412 nmol/mg, the scop 0.5 experimental group was 0.741 nmol/mg, the scop 1 experimental group was 0.815 nmol/mg, and the scop 3 experimental group was 0.789 nmol/mg, and most of the experimental groups had higher AChE than the control group. However, the AChE activity did not increase dose-dependently (Figure 6B).

### 3.7. Neuronal Cell Death Level Effect Depending on Scopolamine Dose

The changes in neuronal cell death according to the dose of scopolamine in the cerebral cortex and hippocampal protein were analyzed using Western blotting. In the cerebral cortex, Bax and Bcl- XL showed a tendency to increase or decrease depending on the dose. BAX increased from 3-fold to 5-fold as the scopolamine dose increased and Bcl- XL decreased by 20–70% (Figure 7C,D). In the hippocampus, Bax decreased in the scop 0.5 group and increased in the scop 3 group, and Bcl-XL decreased in all doses (Figure 7E,F).

These results show that protein expression decreases or increases in a scopolamine-dose-dependent manner in the hippocampus.

## 4. Discussion

AD gradually impairs a patient’s brain function. A common type is memory impairment seen in Alzheimer’s patients. Initially, it is restricted to the patient’s latest experience or piece of learnt information, but gradually older memories are also forgotten. Most of these diseases are closely related to age [41]. Experts predict that dementia will affect more than 10 billion people worldwide by 2050 [42]. Scopolamine contributes to cognitive dysfunction through several mechanisms, such as increasing acetylcholinesterase activity and reducing acetylcholine levels in the synaptic space [43], promoting cytotoxic signaling [44], elevating tau protein levels [45], generating free radicals, inducing oxidative stress [46], and increasing Aβ accumulation [46]. This suggests that scopolamine significantly contributes to the cholinergic hypothesis and can work as an AD research model.

In this work, the scopolamine-induced cognitive dysfunction model was studied with different doses of scopolamine. And the differences in cognitive dysfunction related to marker expression and level in the hippocampus and cortex according to doses of scopolamine were analyzed.

Previous research on scopolamine-induced cognitive dysfunction has not addressed its persistence. In this study, we confirmed that scopolamine-induced cognitive dysfunction persists for 1 h after scopolamine administration (Figure 1). This result was similar to previous studies that also confirmed cognitive dysfunction. In addition to drugs, it can bring a broader treatment approach to scopolamine-induced AD research.

Scopolamine acts as a muscarinic antagonist in the brain, as previously reported. In animals, blocking cholinergic receptors promotes amnesia by impairing spatial learning and memory recognition [47]. In previous studies, spatial learning and memory impairment assessed using the Morris water maze showed a decrease in spatial learning ability, such as an increase in escape time, in the scopolamine-administered group [38,39,48]. However, in our study, we observed a tendency to increase or decrease at doses lower than 1, but we could not confirm statistical significance. Our results confirm that administration of 3 mg/kg scopolamine increased escape time and decreased exploration time in the target quadrant.

The novel object recognition test takes advantage of the characteristic that rodents spend more time exploring novel objects than familiar objects [49]. In several studies, cognitive impairment assessed through the novel object recognition test showed a decrease in the time spent exploring new objects compared to existing objects in the scopolamine administration group, indicating a decrease in cognitive ability [50,51,52]. However, in our study, there was a trend towards an increase or decrease at concentrations lower than 1, but no statistical significance could be identified. When scopolamine (3 mg/kg) was administered, exploration of new objects was found to decrease compared to the control group, which showed that a 3 mg/kg dose of scopolamine caused cognitive impairment. These results are consistent with prior investigations, which found a decrease in the novel object exploration time and recognition index, as well as severe recognition memory impairment with scopolamine at 3 mg/kg.

As shown in Figure 4, we analyzed the in vivo expression of neuromodulator proteins such as BDNF, NGF, and MAP2. Vascular endothelial growth factor, BDNF, and NGF are examples of growth or trophic factors with neurogenesis and neuroprotection. Particularly, BDNF plays a crucial role in neurogenesis, neuroprotection, and synaptic plasticity [53,54]. The brain samples of AD patients show low levels of BDNF mRNA and protein [55,56].

Our study confirmed that the expression of BDNF and NGF decreased in the scop 3 group, which had the highest doses of scopolamine. According to a report by Andrea et al., AD is associated with abnormalities in neurotrophic signaling in the brain, resulting in a gradual decrease in the levels of NGF and BDNF [57]. Shuangshuang et al. showed that a specific extract has the potential to improve cognitive function in Alzheimer’s transgenic mice by regulating BDNF gene expression [58]. Another study demonstrated that scopolamine (2 mg/kg)-treated mice exhibited reduced BDNF levels in the hippocampus compared to the control group [59]. BDNF is involved in the processes that support learning and memory by stabilizing excitatory synapses in the hippocampus and by enhancing their function after stimulation [60].

Our study confirms that administering scopolamine decreased the expression of BDNF in hippocampus, but the 0.5 mg/kg administration group showed an increase in BDNF expression in the cerebral cortex using Western blotting analysis. Also, in the cerebral cortex, it was observed that there was no statistical difference because there was no change in expression after scopolamine administration. Therefore, BDNF analysis must be performed in the hippocampus rather than in the cortex to accurately determine the effect.

Recent research highlights the role of NGF in both aging and age-related conditions such as AD, where age-related disruptions in trophic signaling are linked to the cholinergic and cognitive decline characteristics of AD [61]. And Khushboo et al. demonstrated that administering scopolamine at a dose of 2 mg/kg decreased NGF levels, which is crucial for the survival, maintenance, and development of mammalian neurons in the whole brain [62]. Other researchers have reported that scopolamine (1 mg/kg) administration significantly reduced NGF expression in the hippocampus and cortex [63].

We also investigated the expression of MAP2 in scopolamine-induced animal models in varying doses. MAP2 was chosen due to its prominent presence in the somatodendritic region of the CNS, where it serves as a key indicator of mature neurons and plays a critical role in shaping axonal and dendritic structures within the nervous system [64]. Alterations in MAP2 expression were also reported in animal models of mental disorders, e.g., animals showing cognitive impairment or abnormal behavior relevant to schizophrenic symptoms [65,66]. MAP2 expression is also related to recovery after injury, a remodeling involved in neuronal plasticity or learning [67]. Our study confirmed a scopolamine-dose-dependent alteration in MAP2 expression, suggesting a decrease in microtubule assembly ability as the dose increases. Therefore, it was confirmed through immunohistochemistry that scopolamine administration decreased the expression of BDNF, NGF, and MAP2, and, in particular, through protein analysis, it was confirmed that the expression of BDNF decreased in a dose-dependent manner in the hippocampus more than in the cortex. Also, these results demonstrated neuronal death after scopolamine administration, potentially contributing to the decline in learning ability and memory in MWM and NOR.

Damage to cholinergic neurons is well documented in several neurodegenerative diseases, including AD. Acetylcholine is a neurotransmitter that is widely distributed throughout the nervous system. It is concerned with brain functions like cerebral cortex development, cognitive function, and memory in the central nervous system [68]. The cholinergic system, one of the major neurotransmission pathways, is involved in the mechanisms of memory and cognitive ability [69]. The first pathophysiological component identified in AD is a significant decrease in cholinergic activity [8]. Changes in acetylcholine synthesis or presynaptic recapture accompany the neurodegeneration of cholinergic neurons, leading to a progressive deterioration of memory [28,70,71]. This established the cholinergic hypothesis for the cure of cholinergic deficiency and cognitive recovery in AD patients.

Impairment of cholinergic signaling impairs cognitive function, whereas enhancement of cholinergic signaling improves memory processes. Alzheimer’s patients show cognitive deficits due to the dysfunction of cholinergic neurons [72]. Scopolamine inhibits cholinergic neurons by blocking cholinergic signals in the cerebral cortex and hippocampus, resulting in memory impairment in healthy animals. Scopolamine treatment increased the number of damaged neurons in the hippocampus of male Wistar rats [73]. The administration of scopolamine significantly reduced cholinergic neurotransmitter levels in the whole brains of male BALB/c mice [74]. Acetylcholine is synthesized using choline acetyltransferase, and its action is terminated using AChE and butyrylcholinesterase. Scopolamine treatment increased AChE and butyrylcholinesterase activities in the hippocampus [75], and it also reduced ChaT activity in the hippocampus [76,77].

In our study, AChE activity increased in the cerebral cortex according to a scopolamine-dose-dependent manner. In the hippocampus, activity increased when scopolamine was administered, but activity levels did not appear to increase even as the scopolamine dose increased. So, in order to analyze AChE activity, it can be seen that analyzing the cerebral cortex is more accurate than the hippocampus. Therefore, it is more accurate to analyze AChE activity in the cerebral cortex than in the hippocampus. In addition, our results show that scopolamine administration increased acetylcholinesterase activity. This suggests that acetylcholine release into the synaptic cleft may be reduced and muscarinic activity may be reduced, ultimately leading to decreased cognitive function.

Another hallmark of AD is increased oxidative stress in neurons [78]. In animal models, scopolamine-induced cognitive impairment is associated with changes in the oxidative stress state of the brain. Lipid peroxidation is an oxidative degradation in which free radicals, which are oxidants, attack lipids containing carbon–carbon double bonds [79]. MDA is an organic compound produced by lipid peroxidation and is a biomarker for oxidative stress [80]. It has been shown that scopolamine-induced cognitive decline is related to drastic elevation of MDA levels in mice and rats [81,82,83,84].

Scopolamine administration decreases antioxidant enzymes such as catalase, glutathione peroxidase (GSH-Px), and superoxide dismutase (SOD), as well as antioxidant substrates such as reduced glutathione (GSH). Intraperitoneal long-term injection of scopolamine (3 mg/kg/day) for 14 days decreased SOD levels and increased MDA levels by 2.5-fold in the whole brains of mice [81]. A study in male Kunming mice showed that scopolamine administered at a dose of 2 mg/kg (i.p.) increased MDA levels by 1.8-fold and decreased SOD activity in the hippocampus [82]. In another study, scopolamine treatment (1 mg/kg, i.p.) decreased SOD levels and increased MDA by 3-fold in male BALB/c mice in the hippocampus [85]. Additionally, scopolamine (1 mg/kg, i.p.) administered 30 min before behavioral testing decreased SOD levels and increased MDA activity by 1.4-fold in the hippocampus of male Wistar rats [86]. In our study, MDA levels increased from 2.5- to 4-fold in the cerebral cortex in a scopolamine-dose-dependent manner. But in the hippocampus, MDA levels increased by about 2-fold when scopolamine was administered, but MDA levels did not appear to increase even when the scopolamine dose was increased.

Previously, many researchers performed MDA analyses in the hippocampus. However, as a result of our study, we were able to observe a more-differentiated MDA increase in the cortex, so antioxidant assessments can be analyzed more significantly in the cortex than in the hippocampus.

Apoptosis, or programmed cell death, plays a critical role in tissue homeostasis and normal development. Nevertheless, apoptosis has been identified in neurodegenerative diseases such as AD [87,88]. Apoptosis in hippocampal and cortical neurons contributes to learning and memory impairment [89,90]. Bax is an apoptotic protein, while Bcl-2 is an important anti-apoptotic protein [91]. Bcl-XL is a member of the Bcl-2 protein family which acts as an anti-apoptotic protein, is essential for neuronal survival, and plays a role in protecting against neuronal damage [92]. According to a study, scopolamine was reported to increase Bax expression and decrease Bcl-XL expression in a protein expression analysis in rat hippocampi [93]. In addition, another study using immunostaining reported that scopolamine increased Bax in the whole brains of rats and decreased Bcl-2 protein expression levels [94].

Our results show that the 0.5 mg/kg administration group showed an increase in BDNF expression in the cerebral cortex and a decrease in Bax expression in the hippocampus, likely due to mild nerve damage to the cholinergic system from scopolamine. Overall, Bax increased in a dose-dependent manner in the cortex and Bcl-XL decreased. However, in the hippocampus, Bax decreased at scopolamine doses between 0.5 and 1 mg/kg and increased when 3 mg/kg of scopolamine was administered, and in the case of Bcl-XL, it decreased in the opposite direction to Bax. In our results, the increased rate in Bax and the decreased rate in Bcl-XL were analyzed to be more concentration-dependent in the hippocampus than in the cortex.

## 5. Conclusions

This study verifies that a 3 mg/kg injection of scopolamine in the scopolamine-induced cognitive dysfunction’s mouse model effectively reduced spatial and recognition memory and increased apoptosis.

Our study confirms that administering scopolamine decreased the expression of BDNF in the hippocampus, but the low doses in the administration group showed an increase in BDNF expression in the cerebral cortex after using Western blotting analysis. Also, in the cerebral cortex, it was observed that there was no statistical difference because there was no change in expression after scopolamine administration. Therefore, protein analysis must be performed in the hippocampus rather than in the cortex to accurately determine the effect.

But AChE activity and the MDA level increased in the cerebral cortex in a scopolamine-dose-dependent manner. In the hippocampus, activity increased when scopolamine was administered, but activity levels did not appear to increase even as the scopolamine dose increased. Thus, AChE activity and MDA analysis can lead to more-accurate results in the cerebral cortex than the hippocampus.

Therefore, if hippocampus and cortex analyses are performed separately, depending on the analysis’ purpose and target, the number of specimens can be increased so that an effective research analysis can be derived.

## Figures and Tables

**Figure 1 biomedicines-12-02475-f001:**
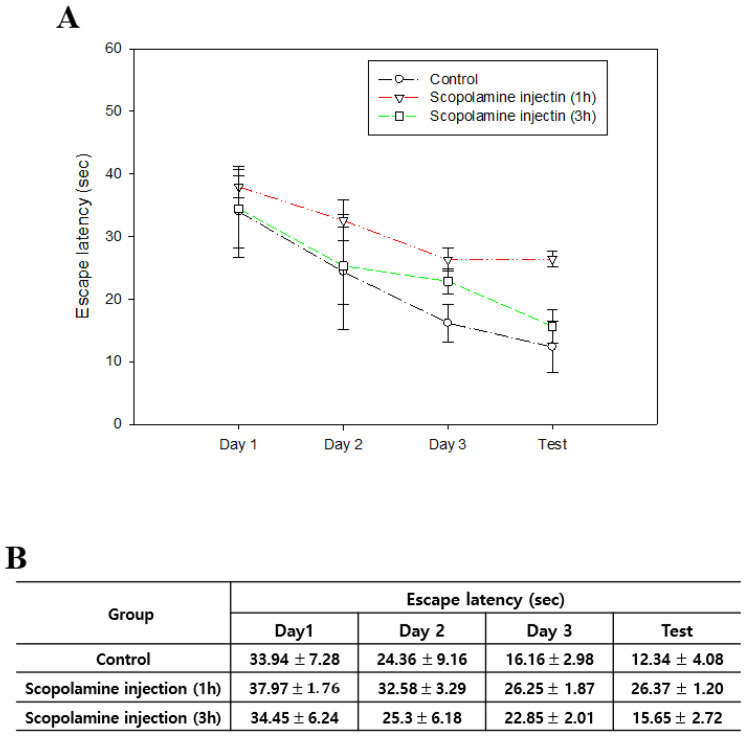
A comparative study of the duration of cognitive impairment induced by scopolamine. Graph (**A**) and measurements (**B**) of escape latency over time following scopolamine administration. The data are presented as the mean ± standard error.

**Figure 2 biomedicines-12-02475-f002:**
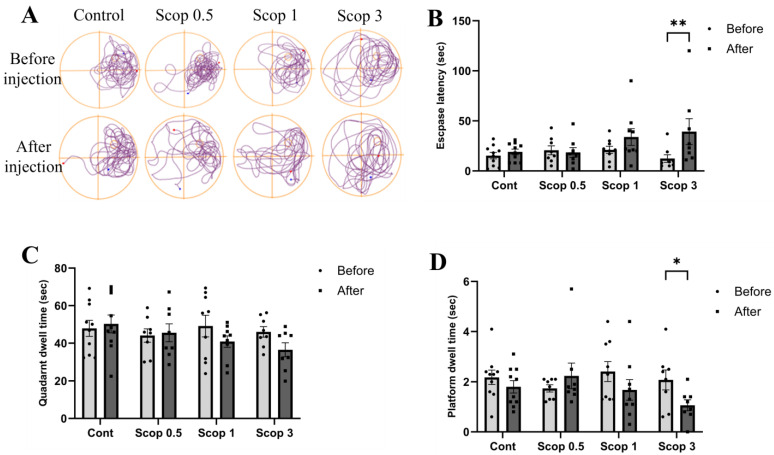
Scopolamine injection induces spatial learning and memory in mice. (**A**) Representative swim paths of rats during the probe test. (**B**) Escape latencies in scopolamine injection before and after. (**C**) Quadrant dwelling time before and after scopolamine injection. (**D**) Platform dwelling time before and after scopolamine injection. n ≥ 8 per group. The symbols * and ** indicate significant differences at *p* < 0.05 and *p* < 0.01, respectively. Scop 0.5: scopolamine 0.5 mg/kg injection group; Scop 1: scopolamine 1 mg/kg injection group; Scop 3: scopolamine 3 mg/kg injection group.

**Figure 3 biomedicines-12-02475-f003:**
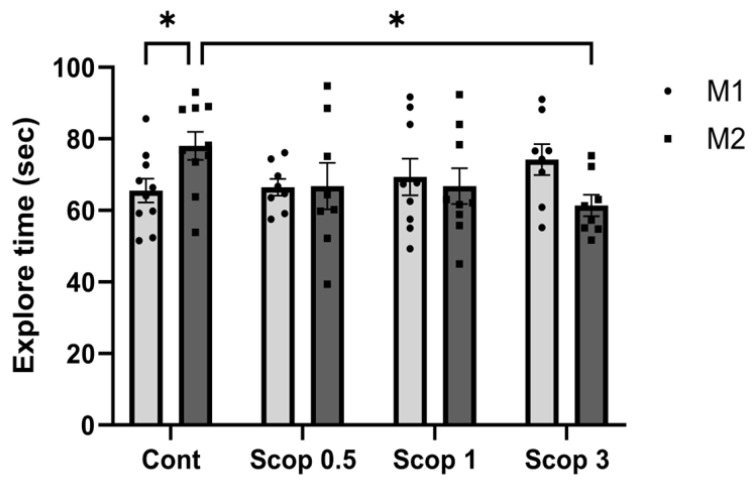
Effect of scopolamine concentration on recognition memory impairment in mice. M1: original object; M2: novel object. Exploration time increased in the control group but decreased in the scop 3 group. n ≥ 8 per group. The symbols * indicate significant differences at *p* < 0.05, respectively. Scop 0.5: scopolamine 0.5 mg/kg injection group; Scop 1: scopolamine 1 mg/kg injection group; Scop 3: scopolamine 3 mg/kg injection group.

**Figure 4 biomedicines-12-02475-f004:**
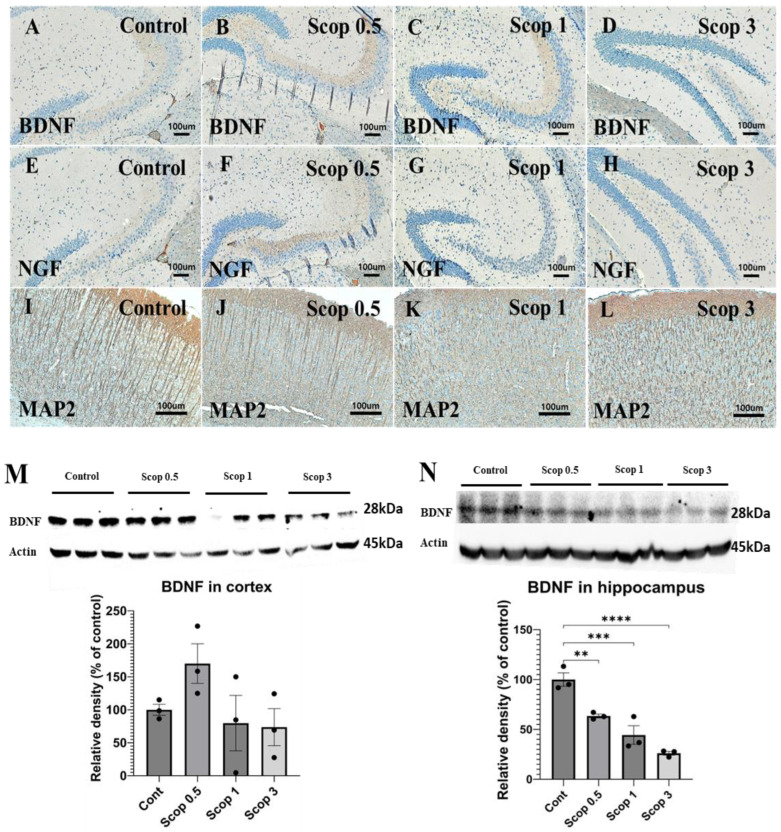
Results of immunohistochemical staining and neurotrophic factor Western blot Showing the expression of BDNF (**A**–**D**) and NGF (**E**–**H**) in the hippocampus, and MAP2 (**I**–**L**) expression in the cortex. (n = 5 per group). Western blot analyses of BDNF in the cortex (**M**) and hippocampus (**N**). BDNF was normalized to beta-actin protein level. Original magnification: (**A**–**H**) ×100, (**I**–**L**) ×200, scale bar = 100 μm. The symbols **, ***, and **** indicate significant differences at *p* < 0.01, *p* < 0.001, and *p* < 0.0001, respectively.

**Figure 5 biomedicines-12-02475-f005:**
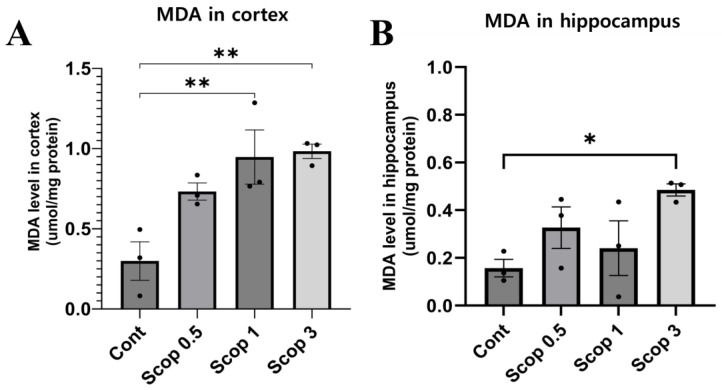
Effects of scopolamine on MDA levels in the cortex (**A**) and hippocampus (**B**). n = 3 per group. The symbols * and ** indicate significant differences at *p* < 0.05 and *p* < 0.01, respectively. Scop 0.5: scopolamine 0.5mg/kg injection group; Scop 1: scopolamine 1 mg/kg injection group; Scop 3: scopolamine 3 mg/kg injection group.

**Figure 6 biomedicines-12-02475-f006:**
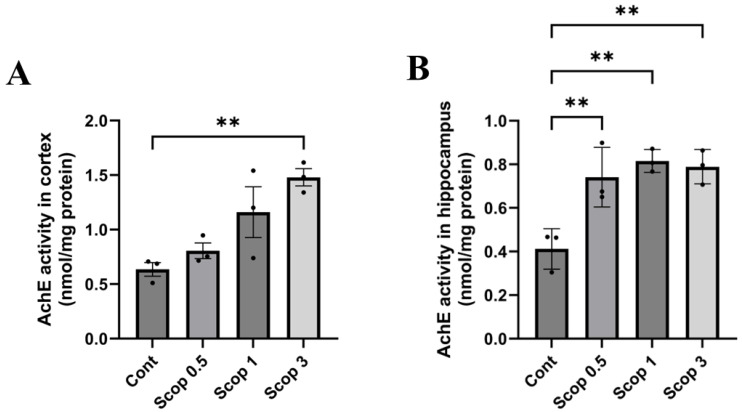
Effects of scopolamine on acetylcholinesterase (AchE) activity in the cortex and hippocampus. AchE activity levels are shown for the cortex (**A**) and hippocampus (**B**). n = 3 per group. The symbols ** indicate significant differences at *p* < 0.01, respectively. Scop 0.5: scopolamine 0.5 mg/kg injection group; Scop 1: scopolamine 1 mg/kg injection group; Scop 3: scopolamine 3 mg/kg injection group.

**Figure 7 biomedicines-12-02475-f007:**
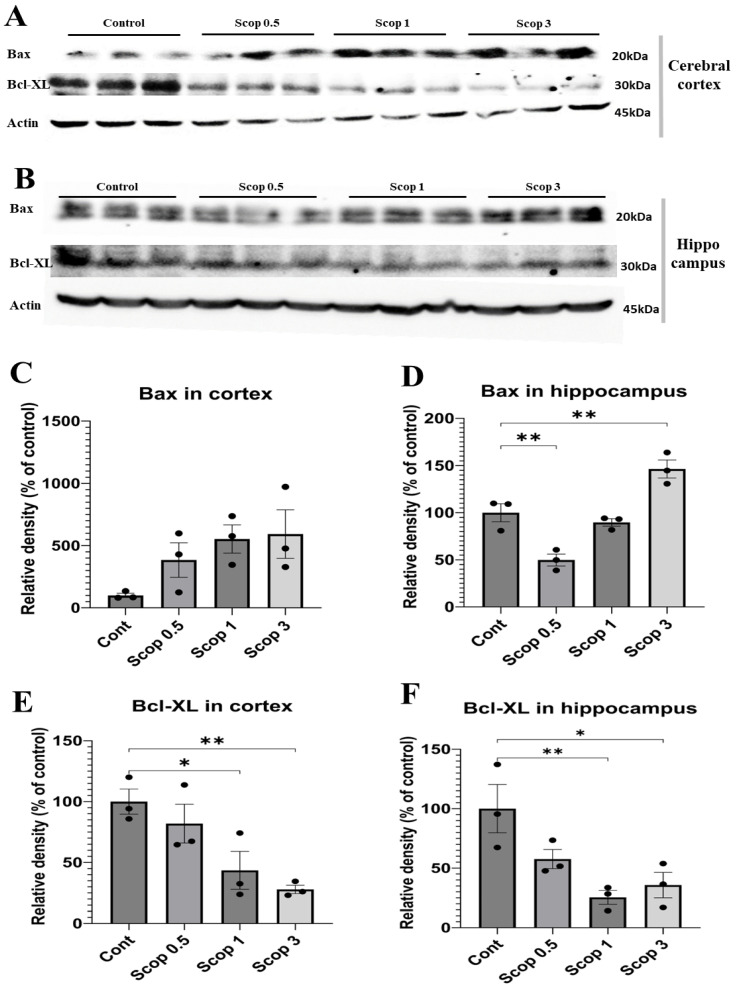
Expression of apoptosis-related markers (Bax, Bcl-XL) in the cortexes and hippocampi used in scopolamine-induced cognitive dysfunction models. Western blot analysis of protein expressions in the cortex (**A**) and hippocampus (**B**). (**C**–**F**) Quantification graph for each protein (% of control). All proteins were normalized to the beta-actin protein level (n = 3 per group). The symbols * and ** indicate significant differences at *p* < 0.05 and *p* < 0.01, respectively. Scop 0.5: scopolamine 0.5 mg/kg injection group; Scop 1: scopolamine 1 mg/kg injection group; Scop 3: scopolamine 3 mg/kg injection group.

**Table 1 biomedicines-12-02475-t001:** Relative staining intensity scores for the expression of BDNF, NGF, and MAP2.

Marker	Control	Scop 0.5	Scop 1	Scop 3
BDNF	+++	+++	+++	+
NGF	+	++	−	−
MAP2	+++	++	−	−

The relative staining intensity was scored as follows: no or weak staining (−), low intensity (+), moderate intensity (++), and strong intensity (+++).

## Data Availability

Data are contained within the article.

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
