# Peer review of "Comparison of Malondialdehyde, Acetylcholinesterase, and Apoptosis-Related Markers in the Cortex and Hippocampus of Cognitively Dysfunctional Mice Induced by Scopolamine"

_biomedicines, 2024, doi:10.3390/biomedicines12112475_

Round 1

Reviewer 1 Report

Comments and Suggestions for Authors

                                                                                                          Date: 08-Oct-2024

Journal: Biomedicines (ISSN 2227-9059)

Manuscript ID: biomedicines-3262105

Type: Article

Title: Comparison of cortex and hippocampus for malondialde-hyde/acetylcholinesterase and brain-derived neurotrophic factor by concentrations of scopolamine in mice.

Authors: 희정 , Myeong-Hyun Nam , Ji-Hoon Park , Ji-Min Lee , Hye-Sun Hong , Tae-Woo Kim , In-Ho Lee , Chang-Ho Shin , Soo-Hong Lee , Young-Kwon Seo *

Brief Summary of the article

This study investigated the effects of different scopolamine concentrations on cognitive function and molecular markers in mice. The researchers found that scopolamine administration decreased behavioral performance in a dose-dependent manner, but this effect was temporary. They also observed changes in oxidative stress markers (MDA) and neurotrophic factors (BDNF) in the brain regions studied. The results suggest that the cerebral cortex and hippocampus may be suitable for different types of analyses to assess the impact of scopolamine on brain function. There are major corrections that need to be answered before publishing the article.

1.     The similarity index (39%) of the article seems to be way high than the limit of 15%. Please work on it and reduce the same.

2.     Please include the ladder molecular weight in the western blot images Fig. 6A, represented in the manuscript to get an idea at what range the membranes were cut. This range should be present in each protein marker. Also beta actin should be normalized. Similarly, please change Fig. 7A actin and associated blots

3.     Figure 8, BDNF and NGF IHC, different regions of the hippocampus have been selected in every section. The authors should consider using the same region of the hippocampus in all the sections to maintain uniformity throughout the data analysis.

4.     Fig. 2, MWM track plots and the associated graphs lack clarity.

5.     The results and discussion sections need to be modified as it lacks clarity and does not link with the data.

Comments on the Quality of English Language

The overall language and description in the results and discussion should be improved.

Author Response

Reviewer 1.

This study investigated the effects of different scopolamine concentrations on cognitive function and molecular markers in mice. The researchers found that scopolamine administration decreased behavioral performance in a dose-dependent manner, but this effect was temporary. They also observed changes in oxidative stress markers (MDA) and neurotrophic factors (BDNF) in the brain regions studied. The results suggest that the cerebral cortex and hippocampus may be suitable for different types of analyses to assess the impact of scopolamine on brain function. There are major corrections that need to be answered before publishing the article.

  1. The similarity index (39%) of the article seems to be way high than the limit of 15%. Please work on it and reduce the same.

Answer) We appreciate your keen point out. We have revised the entire manuscript and attached it.

  1. Please include the ladder molecular weight in the western blot images Fig. 6A, represented in the manuscript to get an idea at what range the membranes were cut. This range should be present in each protein marker. Also beta actin should be normalized. Similarly, please change Fig. 7A actin and associated blots

Answer) We appreciate your keen point out. We have revised the figure and attached the original data.

Original data

  • Western blot original image
  • Bax (20kDa), Bcl-XL (30kDa), BDNF (28kDa), Actin (45kDa)
  1. Figure 8, BDNF and NGF IHC, different regions of the hippocampus have been selected in every section. The authors should consider using the same region of the hippocampus in all the sections to maintain uniformity throughout the data analysis.

Answer) We appreciate your keen point out. We have revised the figure and its results, and incorporated these revisions into the manuscript.

  1. Fig. 2, MWM track plots and the associated graphs lack clarity.

Answer) We appreciate your keen point out. We have revised the graph as follows and incorporated it into the manuscript.

  1. The results and discussion sections need to be modified as it lacks clarity and does not link with the data.

Answer) We appreciate your keen point out. We have completely rewritten the Results and Discussion sections and revised the manuscript accordingly.

Reviewer 2 Report

Comments and Suggestions for Authors

The title is confusing, for example here we are talking about doses and not concentrations of scopolamine!

The Abstract is incomprehensible unless you already know what it is talking about!!

The Introduction is written in a "coarse" way and with statements that are not true: for example that Ach is re-uploaded into the presynaptic space ??.

The acronyms used to indicate scopolamine are completely erratic, arbitrary and changeable in the text; also as in the abstract we still talk about concentrations and not doses. Scopolamine is supposed to be administered intraperitoneally, but it is not specified!!

But in addition to the general confusion with which the paper is written, there are already data in the literature that show these same results in the rat even if in a slightly different way (see for example doi.org/10.29252%2FNIRP.BCN.9.1.5 or DOI 10.1007/s11064-015-1780-1).

The authors mention here Alzheimer's or scopolamine-induced dementia, but this method is not appropriate for inducing a chronic-degenerative pathology like those. It is a test for "transient amnesia disease", but this as a specific disorder is never mentioned in this manuscript.

I do not find this study innovative and in any case the data described and presented are chaotic and do not follow a rational path throughout the manuscript.

Comments on the Quality of English Language

It is necessary to revise the entire manuscript

Author Response

Reviewer 2.

1.The title is confusing, for example here we are talking about doses and not concentrations of scopolamine! The Abstract is incomprehensible unless you already know what it is talking about!!

Answer) We appreciate your keen point out. We have revised the title and abstract as follows and incorporated it into the manuscript.

Title

Comparison of cortex and hippocampus for malondialdehyde, acetylcholinesterase and apoptosis related marker in cognitive dysfunction mice induced by scopolamine.

Abstract

Objectives: Until now, many researchers have conducted evaluation on hippocampus for analysis for cognitive dysfunction's model using scopolamine. However, depending on the purpose of analysis, there are differences in the experimental results for hippocampus and cortex. Therefore, this study intends to compare various analyzes of cognitive dysfunction after scopolamine administration with each other in hippocampus and cortex. Methods: Scopolamine was administered at three dosages in mice: 0.5, 1, and 3 mg/kg. And study evaluated differences in cognitive function and the expression of malondialdehyde (MDA), acetylcholinesterase (AChE), and brain-derived neurotrophic factor (BDNF) in the hippocampus and cortex based on scopolamine dosages Results: The Morris water maze test was conducted 1 and 3 hours after scopolamine injection to assess its duration. A significant decrease in behavioral ability was evaluated at 1 hour and observed a similar recovery to the normal group at 3 hours. And Morris water maze escape latency showed differences depending on scopolamine concentration. While the escape waiting time in the control group and scop 0.5 administration group remained similar to before administration, the administration of scop 1 and 3 increased it. In the experimental group administered scop 1 and 3, cerebral MDA levels in the cerebral cortex significantly increased. In the hippocampus, the MDA level of scopolamine-administered groups slightly increased than the cortex. Western blotting assay showed that Bax and Bcl-xl showed a tendency to increase or decrease depending on the concentration, but BDNF increased in scop 0.5, and scop 1 and 3 did not show a significant decrease than the control at the cerebral cortex. In the hippocampus, BDNF showed a concentration-dependent decrease in expression. Conclusion: The study's findings indicate that chemical analyses for MDA and AChE can be performed in the cerebral cortex, while the hippocampus is better suited for protein analysis for apoptosis and BDNF.

2.The Introduction is written in a "coarse" way and with statements that are not true: for example that Ach is re-uploaded into the presynaptic space?

Answer) We appreciate your keen point out. We have revised the introduction as follows and incorporated it into the manuscript.

  1. Introduction

Neurodegenerative diseases are chronic conditions that impair or destroy the nervous system, especially the brain, over time. Most of these conditions are closely related to age [1, 2].

Alzheimer's disease (AD) is a major neurodegenerative disorder that presents with progressive memory loss and cognitive decline, finally leading to respiratory distress and death [3, 4]. The pathological characteristic of AD is amyloid-beta (Aβ) plaque formation [5, 6]. Accumulation of Aβ plaques causes degeneration and death of cholinergic neurons, resulting in synapse loss, which causes cognitive dysfunction and ultimately death [3, 4, 7-9]. Therefore, acetylcholine plays an integral role in the treatment and alleviation of AD [10-14]. Computer simulations have explored many new drug candidates over the past few years, but the clinical development stage has confirmed their ineffectiveness [15, 16]. Selecting an animal model suitable for the new drug development process is an important step in the process [17]. Since AD involves several interconnected molecular pathways, appropriate animal models must be created to understand the underlying mechanisms and develop new drugs [6, 18, 19]. Cognitive decline, behavioral imbalance, deterioration of neuronal function, neuronal loss, and oxidative stress are common in AD that are caused by aging or drugs [20-22]. Also, many studies have shown that BDNF plays a crucial role in neurodegenerative disorders [23-27].

Scopolamine is a competitive antagonist of muscarinic acetylcholine receptors that blocks cholinergic neurotransmission and causes memory impairment in rodents [28-30]. Recent studies have shown that scopolamine causes oxidative stress, leading to amassing the reactive oxygen species and memory impairment [31, 32]. Additionally, Scopolamine can induce apoptosis in hippocampus neuronal cells [33, 34]. The cholinergic hypothesis may be applicable, with scopolamine injections causing cognitive and memory deficits like those observed in AD. Through the development of transgenic mouse models, AD models such as the 5X FAD model and the APP/PS1 model have been developed, but their prices are high. In particular, there is a problem that some models show normal levels in exercise performance evaluations although transgenicity is confirmed. Therefore, it can be a financial burden in experiments evaluating many animals.

When creating a specific disease model, the clinical phenotype characteristics must be considered. In related studies to creating a congnitive dysfuntion model using scopolamine, various doses have been administered [35-38]. These different doses of scopolamine show differences in model formation, which can lead to different research results. Related researchers compared single and repeated administrations of scopolamine at doses of 0.5, 1, and 3 mg/kg, and compared the cognitive decline resulting from this through a memory retrieval test. In addition, hippocampus tissue was analyzed for AChE and MDA. As a result, it was reported that AChE and MDA increased with single administration, and the effect increased further with repeated administration [39].

Analysis of cognitive dysfunction requires mRNA and protein analysis for various markers in addition to chemical analysis. Many researchers have conducted evaluations targeting the hippocampus for analysis. However, there has been no study that performed a cross-comparison evaluation of the hippocampus and cortex for these various analyses. Therefore, this study aims to cross-comparison various analyses of cognitive dysfunction after scopolamine administration in the hippocampus and cortex.

To conduct research on cognitive dysfunction based on changes in exploratory behavior (i.e., novel object recognition test), learning and memory behavior (i.e., Morris water maze), and cholinergic system in a new environment according to the doses of scopolamine (0.5, 1, and 3 mg/kg), and analysis biomarker related to cognitive function, such as apoptotic marker, MDA levels, AChE activity and BDNF expression in the hippocampus and cortex.

3.The acronyms used to indicate scopolamine are completely erratic, arbitrary and changeable in the text; also as in the abstract we still talk about concentrations and not doses. Scopolamine is supposed to be administered intraperitoneally, but it is not specified!!

Answer) We appreciate your keen point out. We have revised the material & methods as follows and incorporated it into the manuscript.

  1. Materials and Methods

2.2.2. Design of scopolamine dose experiment

Scopolamine was set at three doses: 0.5, 1, and 3 mg/kg. The four groups were the following: (1) saline (control); (2) 0.5 mg/kg scopolamine (scop 0.5), (3) 1 mg/kg scopolamine (scop 1), and (4) 3 mg/kg scopolamine (scop 3). The mice conducted the behavior tests 1 hour after scopolamine intraperitoneal injection. All mice performed the Morris water maze test and the novel object recognition test. One day after the end of the preceding behavioral experiments, the mice were administered scopolamine before being sacrificed for biochemical analyses, Western blot, and immunohistochemistry (IHC).

  1. But in addition to the general confusion with which the paper is written, there are already data in the literature that show these same results in the rat even if in a slightly different way (see for example doi.org/10.29252%2FNIRP.BCN.9.1.5 or DOI 10.1007/s11064-015-1780-1).
  2. The authors mention here Alzheimer's or scopolamine-induced dementia, but this method is not appropriate for inducing a chronic-degenerative pathology like those. It is a test for "transient amnesia disease", but this as a specific disorder is never mentioned in this manuscript.
  3. I do not find this study innovative and in any case the data described and presented are chaotic and do not follow a rational path throughout the manuscript.

Answer) We appreciate your keen point out. We have revised the abstract, introduction and conclusion in response to Questions 4 to 6 as follows and incorporated it into the manuscript.

Abstract

Objectives: Until now, many researchers have conducted evaluation on hippocampus for analysis for cognitive dysfunction's model using scopolamine. However, depending on the purpose of analysis, there are differences in the experimental results for hippocampus and cortex. Therefore, this study intends to compare various analyzes of cognitive dysfunction after scopolamine administration with each other in hippocampus and cortex. Methods: Scopolamine was administered at three dosages in mice: 0.5, 1, and 3 mg/kg. And study evaluated differences in cognitive function and the expression of malondialdehyde (MDA), acetylcholinesterase (AChE), and brain-derived neurotrophic factor (BDNF) in the hippocampus and cortex based on scopolamine dosages

Introduction

When creating a specific disease model, the clinical phenotype characteristics must be considered. In related studies to creating a congnitive dysfuntion model using scopolamine, various doses have been administered [35-38]. These different doses of scopolamine show differences in model formation, which can lead to different research results. Related researchers compared single and repeated administrations of scopolamine at doses of 0.5, 1, and 3 mg/kg, and compared the cognitive decline resulting from this through a memory retrieval test. In addition, hippocampus tissue was analyzed for AChE and MDA. As a result, it was reported that AChE and MDA increased with single administration, and the effect increased further with repeated administration [39].

Analysis of cognitive dysfunction requires mRNA and protein analysis for various markers in addition to chemical analysis. Many researchers have conducted evaluations targeting the hippocampus for analysis. However, there has been no study that performed a cross-comparison evaluation of the hippocampus and cortex for these various analyses. Therefore, this study aims to cross-comparison various analyses of cognitive dysfunction after scopolamine administration in the hippocampus and cortex.

To conduct research on cognitive dysfunction based on changes in exploratory behavior (i.e., novel object recognition test), learning and memory behavior (i.e., Morris water maze), and cholinergic system in a new environment according to the doses of scopolamine (0.5, 1, and 3 mg/kg), and analysis biomarker related to cognitive function, such as apoptotic marker, MDA levels, AChE activity and BDNF expression in the hippocampus and cortex.

Conclusion

This study verified that a 3 mg/kg injection of scopolamine in the scopolamine-induced cognitive dysfunction's mouse model effectively reduced spatial and recognition memory and increased apoptosis.

Our study confirmed that administering of scopolamine decreased the expression of BDNF in hippocampus, but the low-doses of administration group showed an increase in BDNF expression in the cerebral cortex by western blotting analysis. Also, in the cerebral cortex, it was observed that there was no statistical difference because there was no change in expression after scopolamine administration. Therefore, protein analysis must be performed in hippocampus rather than in cortex to accurately determine the effect.

But, AChE activity and MDA level increased in the cerebral cortex in a scopolamine doses-dependent manner. In the hippocampus, activity increased when scopolamine was administered, but activity levels did not appear to increase even as the scopolamine dose increased. Thus, AChE activity and MDA analysis can lead to more accurate results in the cerebral cortex than the hippocampus. Therefore, if hippocampus and cortex analysis are performed separately depending on the analysis purpose and target, the number of specimens can be increased, so effective research analysis can be derived.

Round 2

Reviewer 1 Report

Comments and Suggestions for Authors

The manuscript can be accepted. I'm satisfied with the reviewer's response and the corrections in the manuscript.

Author Response

Reviewer 1

Comments and Suggestions for Authors

The manuscript can be accepted. I'm satisfied with the reviewer's response and the corrections in the manuscript.

Answer) Thank you very much for accepting my manuscript. I appreciate your valuable feedback and insights during the review process, which have greatly contributed to improving the quality of the work.

Reviewer 2 Report

Comments and Suggestions for Authors

The authors have largely satisfied my criticisms in re-setting the entire manuscript. Before publication, I suggest mentioning an aspect that is almost never mentioned when using the scopolamine as memory deficit model. I refer to a type of memory deficit, which is not debilitating like classic dementias but which is reproduced very precisely by scopolamine in acute, it is Transient Global Amnesia (TGA). It is a benign and transient anterograde amnesia that regresses in 24 hours in which the muscarinic deficit also affects the central nor-adrenergic tone (see and cite Belardo et al, Int. J. Mol. Sci. 2023).

Author Response

Reviewer 2

Comments and Suggestions for Authors

The authors have largely satisfied my criticisms in re-setting the entire manuscript. Before publication, I suggest mentioning an aspect that is almost never mentioned when using the scopolamine as memory deficit model. I refer to a type of memory deficit, which is not debilitating like classic dementias but which is reproduced very precisely by scopolamine in acute, it is Transient Global Amnesia (TGA). It is a benign and transient anterograde amnesia that regresses in 24 hours in which the muscarinic deficit also affects the central nor-adrenergic tone (see and cite Belardo et al, Int. J. Mol. Sci. 2023).

Answer) We appreciate your keen point out. Following your advice, we have revised the introduction section of the manuscript as follows.

  1. Introduction

“Scopolamine is a competitive antagonist of muscarinic acetylcholine receptors that blocks cholinergic neurotransmission and causes memory impairment in rodents . Recent studies have shown that scopolamine causes oxidative stress, leading to amassing the reactive oxygen species and memory impairment . Additionally, Scopolamine can induce apoptosis in hippocampus neuronal cells . The cholinergic hypothesis may be applicable, with scopolamine injections causing cognitive and memory deficits like those observed in AD.

Additionally, scopolamine has been known to precisely replicate a condition that, while not as severe as classic dementias, manifests as acute Transient Global Amnesia (TGA). This is a benign and transient form of anterograde am-nesia that typically resolves within 24 hours, which was induced by the muscarinic deficit influences central noradrenergic tone. [35]

Through the development of transgenic mouse models, AD models such as the 5X FAD model and the APP/PS1 model have been developed, but their prices are high. In particular, there is a problem that some models show normal levels in exercise performance evaluations although transgenicity is confirmed. Therefore, it can be a financial burden in experiments evaluating many animals.“

References

  1. Belardo, C.; Boccella, S.; Perrone, M.; Fusco, A.; Morace, A.M.; Ricciardi, F.; Bonsale, R.; Elbini-Dhouib, I.; Guida, F.; Luongo, L.; Bagetta, G.; Scuteri, D.; Maione, S., Scopolamine-Induced Memory Impairment in Mice: Effects of PEA-OXA on Memory Retrieval and Hippocampal LTP. In International Journal of Molecular Sciences, 2023; Vol. 24.
